# GraphARC: A Comprehensive Benchmark for Graph-Based Abstract Reasoning

## Abstract

Relational reasoning lies at the heart of intelligence, but existing benchmarks are typically confined to formats such as grids or text. We introduce *GraphARC*, a benchmark for abstract reasoning on graph-structured data. GraphARC generalizes the few-shot transformation learning paradigm of the Abstraction and Reasoning Corpus (ARC). Each task requires inferring a transformation rule from a few input-output pairs and applying it to a new test graph, covering local, global, and hierarchical graph transformations. Unlike grid-based ARC, GraphARC instances can be generated at scale across diverse graph families and sizes, enabling systematic evaluation of generalization abilities.

We evaluate state-of-the-art language models on GraphARC and observe clear limitations. Models can answer questions about graph properties but often fail to solve the full graph transformation task, revealing a comprehension-execution gap. Performance further degrades on larger instances, exposing scaling barriers. More broadly, by combining aspects of node classification, link prediction, and graph generation within a single framework, GraphARC provides a promising testbed for future graph foundation models.

## 1 Introduction

Relational reasoning—the ability to perceive and reason about relationships between objects—is a core aspect of intelligence (Hummel & Holyoak, 2003; Halford et al., 2010). This capacity underlies many forms of higher cognition: we use it to appreciate analogies across different domains (Holyoak, 2012), to understand and learn language (Pinker, 1998), and generally, to apply abstract rules to novel situations (Smith et al., 1992). Achieving this kind of generalization is a central challenge for artificial intelligence (Chollet, 2019; Lake et al., 2017).

The Abstraction and Reasoning Corpus (ARC) (Chollet, 2019) is a widely recognized benchmark for evaluating abstract reasoning in AI. ARC consists of grid-based visual puzzles that require systems to learn and apply transformation rules based on a few examples. While ARC puzzles are grid-based, many of the underlying rules are relational—grouping identical objects, replicating subpatterns, or propagating attributes to neighboring objects. To capture such a structure more directly, we propose to use graphs as a more general representation that is not tied to a particular spatial layout.

Inspired by ARC, we introduce *GraphARC*, a benchmark for few-shot abstract reasoning on graphs. Each task includes 2-3 input-output graph pairs demonstrating a transformation rule, and a test input where the rule should be applied. See Figure 1 for an example. The transformations are based on fundamental graph primitives: local structure (degree, neighborhoods, cliques), reachability (connected components, isolated nodes), or hierarchical relations (closest common ancestor in a tree). The transformations can be color-based (changing the color of some nodes) and structural modifications, adding or removing nodes and edges. The instances are automatically generated across diverse graph families and sizes, providing a virtually unlimited supply of instances. This setup allows systematic evaluation and, crucially, tests whether models can generalize the same transformation across graphs of varying size and structure.

In a sense, GraphARC combines elements of traditional graph learning tasks, including node classification (Kipf, 2016), graph classification (Ying et al., 2018), link prediction (Zhang & Chen, 2018), and graph generation (Simonovsky & Komodakis, 2018) within a single framework. Given the absence of broadly applicable Graph Foundation Models (GFMs) (Liu et al., 2025; Wang et al., 2025),

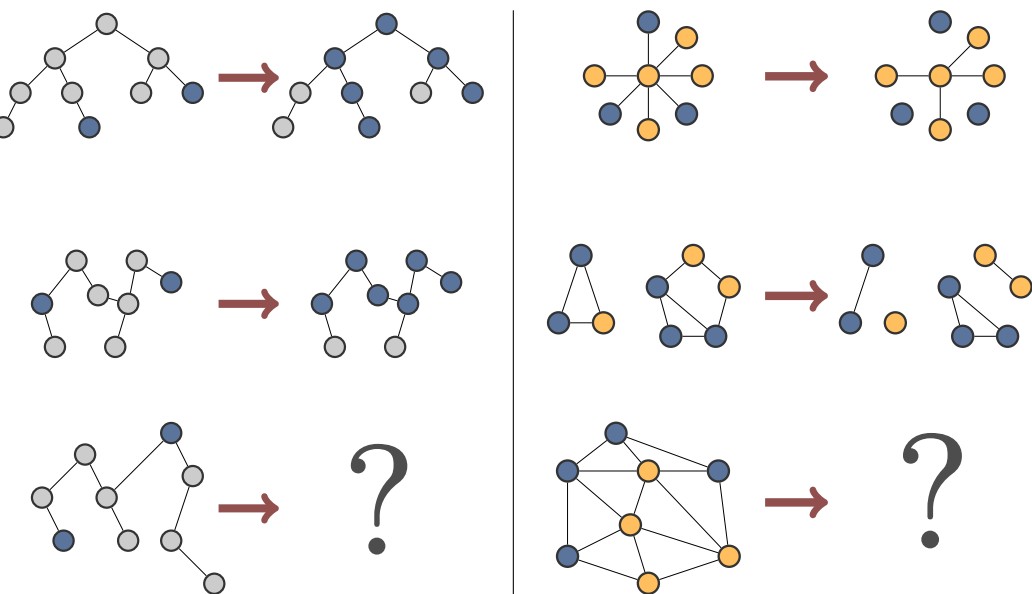

Figure 1: Two examples of GraphARC tasks demonstrating few-shot abstract reasoning on graphs. Each task presents two input-output graph pairs (top row and middle row) that illustrate a transformation rule, followed by a test input graph where the learned rule must be applied. In the example on the left hand side, the transformation rule colors the shortest path connecting two colored nodes (shown in blue). This task is constrained to tree structures to ensure a unique shortest path between any two nodes. The example on the right hand side shows a task where the task is to remove edges between nodes of different colors.

we focus on evaluating Large Language Models (LLMs) that can process textual representations of graphs. To this end, we systematically test different encoding schemes and prompt variations, and we evaluate models on full graph transformation tasks and on targeted questions about properties of the output graph.

Concretely, GraphARC provides

1. a scalable task generation framework that produces diverse graph transformation challenges across multiple graph families at arbitrary scales,

2. a comprehensive evaluation methodology for LLMs, and

3. an extensive analysis of current LLM performance.

This analysis reveals significant gaps in structural understanding and identifies key failure modes that highlight directions for future research.

## 2 RELATED WORK

**The Abstraction and Reasoning Corpus.** The ARC challenge, introduced by Chollet (2019), established a paradigm for few-shot abstract reasoning in which systems infer transformation rules from minimal input–output examples and apply them to new cases. Despite intensive effort, performance hovered near 33% for years before recent breakthroughs: top open-source entries in the ARC Prize combined neural reasoning with programmatic search to reach roughly 53% accuracy. At the same time, OpenAI's o3 attained 75.7% on ARC-AGI-1 (ARC Prize Foundation, 2024b;a). However, the release of ARC-AGI-2 (Chollet et al., 2025) in March 2025 largely reset the field, with frontier systems scoring about 16% versus human performance near 60%, underscoring that robust abstract reasoning remains open. Crucially for our setting, leading ARC solvers are tailored to grid-based vision puzzles and often rely on specialized pipelines or substantial compute, making direct application to GraphARC's relational, graph-structured I/O unnecessary or impractical.

Accordingly, we evaluate general-purpose LLMs on GraphARC's textual graph encodings to probe abstract relational reasoning without bespoke ARC-specific machinery.

The impact of ARC has led to several extensions exploring different modalities and task formulations. (Xu et al., 2024) created 1D-ARC tasks specifically for language model evaluation. Assouel et al. (2022) developed Arith-MNIST, containing reasoning tasks where models must infer arithmetic programs applied to colored digits. In this challenge, the output is a single digit containing the answer, rather than a transformed grid.

**Large Language Models and Graph Reasoning.**   Recent work has explored applying LLMs to graph reasoning tasks with mixed results. Fatemi et al. (2023) explores various textual representations and their impact on LLM performance across different reasoning tasks. Wang et al. (2023) examines whether language models can solve graph problems such as connectivity, cycle existence, and bipartite matching, when graphs are described in text.

Zhang et al. (2024) evaluate how LLMs understand graph patterns through tasks such as detection, translation, and modification, with patterns specified in natural language or as edge lists. This work is arguably the most similar to our work, but it focuses on reasoning about predefined motifs, whereas we target few-shot learning and the application of general graph transformations. Beyond inputting graphs in text, Zhao et al. (2023) introduces GIMLET, using a customized positional encoding to integrate language models with graph-structured data for molecule property prediction. Sanford et al. (2024) investigates how well transformers can solve graph-based reasoning problems at various model sizes. While they do not use natural language to represent the graphs, they do show that transformers can solve graph problems they are trained on. See (Jin et al., 2024) for a comprehensive survey on LLMs for graph problems.

Chain-of-thought prompting (or test time compute) has emerged as an effective way to improve reasoning abilities in language models (Wei et al., 2022; Snell et al., 2025; Mirtaheri et al., 2025). We will investigate this by using closed- and open-source models with reasoning abilities, such as OpenAI's o3-mini and DeepSeek's R1.

**Graph Foundation Models.**   GFMs are an emerging class of models aimed at general-purpose graph reasoning. These approaches seek to combine GNN-style structural inductive biases with the broad knowledge and few-shot learning of foundation models (Liu et al., 2023). Initial efforts have targeted specific domains, such as knowledge graphs (Galkin et al., 2024) and molecular graphs (Méndez-Lucio et al., 2024). See (Liu et al., 2025) for a survey outlining the opportunities and challenges.

Notably, Google Research has announced a relational GFM that treats databases as graphs and generalizes to unseen tables, reporting up to 40x precision gains on tasks like spam detection. Despite rapid progress, current GFMs largely target standard supervised objectives (node/graph classification, link prediction) (Liu et al., 2023; 2025; Wang et al., 2025), and cannot be straightforwardly adapted to solve ARC style questions. Accordingly, we do not benchmark GFMs here; instead, GraphARC serves as a complementary proving ground for future GFMs that claim abstract, few-shot graph transformation capabilities.

## 3   GraphARC

### 3.1   Benchmark Definition

A GraphARC task consists of 2-3 input-output graph pairs that demonstrate a transformation rule. This is followed by a test input graph, where the learned rule must be applied. See Figure 1 for an example task.

Formally, graphs are represented as a tuple $G = (V, E)$ where nodes have unique integer identifiers from 0 to $|V| - 1$. Node IDs remain consistent within each task instance. Each node $v \in V$ carries a color attribute, with grey as the default color. We assume the graphs to be undirected for simplicity.

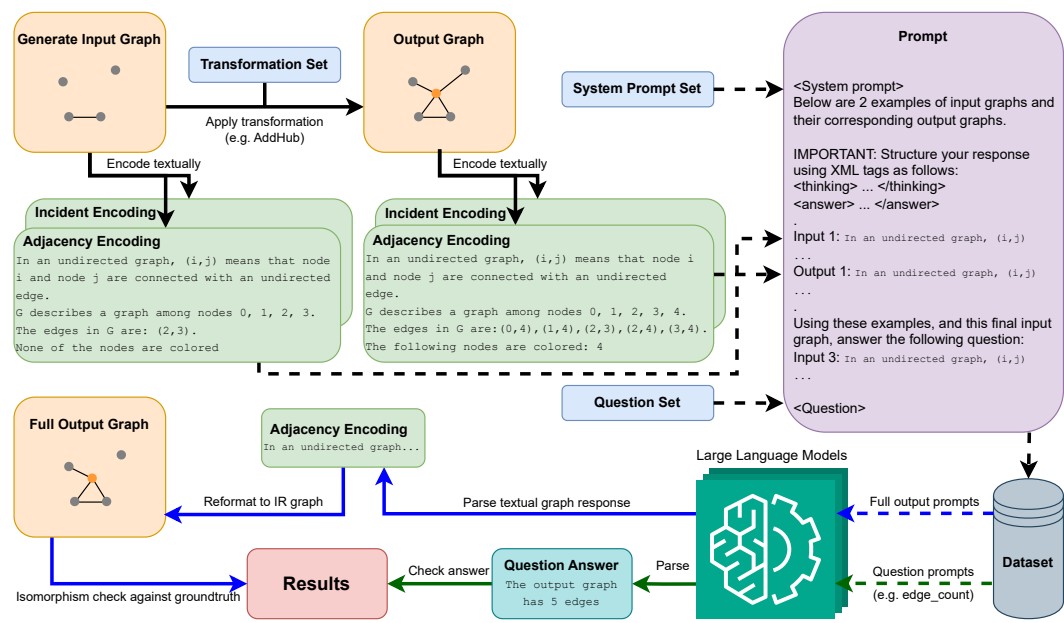

Figure 2: The GraphARC task generation and evaluation pipeline. A task begins with a formal rule (e.g., "color all degree-3 nodes blue"), which is used to generate and textually encode a set of input-output graph examples. The pipeline then diverges into two distinct evaluation pathways: 1) The **Full Output Path (blue arrows)** requires the model to generate the complete transformed graph, testing both its ability to infer the rule and execute it correctly. 2) The **Question-Based Path (green arrows)** isolates comprehension by asking targeted questions about the input graph and the inferred output properties, disentangling understanding from execution. Solid arrows represent stages where analysis or transformations occur, while dashed arrows indicate the flow of data without modification.

**Task categories.** A task is an instance of a transformation, since we can generate multiple tasks per transformation. GraphARC contains 21 distinct transformations. We organize GraphARC transformations into two main categories: color-based tasks and structure modification tasks.

*Color-based tasks* modify node colors based on structural properties, for example:

- `colorDegreeX`: Color all nodes with degree X
- `colorNeighbors`: Color all neighbors of red nodes blue
- `colorComponents`: Extend the color of a node to its entire connected component
- `colorEquidistant`: Color all nodes equidistant from two colored nodes
- `colorPath`: Color the unique shortest path between two colored nodes

*Structure Modification Tasks* alter the graph topology, for example:

- `addHub`: Add a new colored hub node connected to all existing nodes
- `removeDegreeX`: Remove all nodes with degree X
- `bipartitionCompletion`: Complete bipartite coloring from seed nodes
- `removeSameColorEdges`: Remove all edges between nodes of the same color

See Appendix B for a full list and detailed descriptions.

**Graph generation and validation.** We generate graphs from multiple families, including Erdős-Rényi, Watts-Strogatz, trees, star, bipartite, and multi-component, and systematically vary the input graph size $n$ (node count). For structure-modifying transformations, $n$ denotes the *pre-transformation* input size. A property-validation system ensures graph-transformation compatibility using a three-state framework (True/False/Maybe) indicating whether a generator guarantees,

precludes, or conditionally satisfies the required properties, enabling efficient rejection of invalid instances and regeneration only when necessary. Full details appear in Appendix C.

## 3.2 LLM Evaluation Pipeline

Figure 2 illustrates our task generation and evaluation pipeline. Each task begins with a transformation rule used to generate a set of input-output graph examples. These examples are then encoded into text using the adjacency or incident list formats. The pipeline then diverges into two distinct evaluation pathways called the full output prompt and question-based prompt, as detailed in the figure caption. This enables us to distinguish between a model's ability to understand a transformation and its capability to execute it. For instance, a model might correctly answer a question about the output graph's properties (demonstrating understanding) while failing to generate the full, correct graph structure (revealing execution limitations). We also test the effect of different system prompts. See Appendix E for a comprehensive description of these variations, their design rationale, and an analysis of their empirical effects. We instantiate combinations only when the task-generator validator deems them compatible (e.g., `addHub` is valid for random/connected/tree/star, whereas `bipartitionCompletion` is valid only for connected bipartite), which induces different instance counts per transformation.

## 4 Experiments and Results

### 4.1 Dataset and Models

We evaluate a range of SOTA AI systems to provide comprehensive coverage of reasoning capabilities, including Qwen3 (1.7B to 32B), DeepSeek R1, OpenAI's reasoning models o1-mini, o3-mini, o4-mini, and GPT 5 (all with medium reasoning effort), and GPT 4.1-nano as a direct-answer baseline.

There are two regimes in the dataset: (i) our *main experiments*, which span a range of tasks under multiple prompts and encoding variations, and (ii) *scaling experiments*, which assess performance as graph size increases. In the main experiments, we use small graphs, $n \in \{5, 10, 15\}$ for both the example and test graphs. In the scaling experiments, examples are fixed at $n = 10$, while test inputs range up to $n = 250$ with $n \in \{10, 25, 50, 100, 250\}$.

The total number of evaluations per model results from combining different encodings, prompt variants, size patterns, transformations, and question types. After filtering by task-generator compatibility, for Qwen3 models and GPT 4.1-nano, this gives around 18k evaluations each, while reasoning models—restricted to a single prompt—cover a proportionally smaller set. The scaling dataset adds 4.4k evaluations for reasoning models.

### 4.2 Results

Figure 3 reveals a stark performance hierarchy in full graph generation tasks. The Qwen3 series shows clear scaling with model size (8.3% for 1.7B to 41.4% for 14B), while reasoning models achieve substantially higher performance (57.5% for o1-mini to 90.9% for GPT 5). GPT 4.1-nano achieves only 13.3% accuracy, performing similarly to the smallest Qwen3 model.

Models are substantially better in question-answering tasks compared to full graph generation. This gap is most pronounced in smaller models, `Qwen3 1.7B` achieves 54.4% on questions but only 8.3% on full outputs (6.55x difference), while `Qwen3 8B` shows 72.2% vs 30.5% (2.36x difference). Even frontier models exhibit this pattern: o4-mini achieves 91.2% on questions versus 86.5% on full outputs. Models consistently perform better on questions about input graphs (which they can see) than output graphs (which they must infer). For example, `Qwen3 8B` has an accuracy of 89.2% on input questions versus 65.1% on output questions.

**Task-Specific Performance.** Table 1 reveals pronounced variation across tasks. Some transformations, such as `addHub`, `colorDegree1`, and `colorNeighbors`, are consistently solved with high accuracy. In contrast, others, including `removeDegree3`, `colorEquidistant`, and `colorDegree3`, remain challenging even for the strongest models. Global transformations

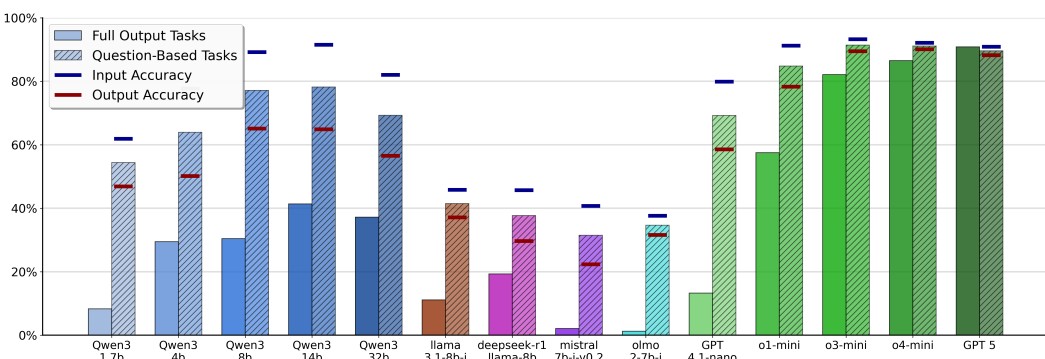

Figure 3: Overall model performance on GraphARC. Bars show accuracy for full-output tasks (structured graph generation) and for question-based tasks, split into questions about the *input* (visible graph) and the *output* (inferred graph). For most models, we observe a comprehension–execution gap (question-based > full-output) and an input–output asymmetry (input > output). Reasoning models consistently outperform direct-answer baselines.

Table 1: Accuracy on full-output tasks across a subset of the tested models. Values are mean accuracies, and bold highlights the best score per row. Tasks are ordered by increasing average difficulty. A description of each task can be found in Appendix B. Appendix H shows results for all models.

| Task | Qwen3 14b | LLaMA3.1 8b | DeepSeek-R1 LLaMA-8b | GPT 4.1-nano | o1 mini | o3 mini | GPT 5 | Avg |
|---|---|---|---|---|---|---|---|---|
| removeDegree3 | 0.02 | 0.00 | 0.00 | 0.00 | 0.00 | 0.25 | **0.75** | 0.15 |
| colorEquidistant | 0.00 | 0.00 | 0.00 | 0.00 | 0.38 | 0.50 | **0.75** | 0.23 |
| colorDegree3 | 0.29 | 0.00 | 0.10 | 0.00 | 0.00 | **0.67** | 0.58 | 0.24 |
| colorDistanceAtLeast2 | 0.00 | 0.00 | 0.00 | 0.00 | 0.08 | 0.83 | **0.92** | 0.26 |
| removeDegree2 | 0.00 | 0.00 | 0.00 | 0.00 | 0.42 | 0.75 | **1.00** | 0.31 |
| colorMaxDegree | 0.52 | 0.25 | 0.04 | 0.25 | 0.25 | 0.33 | **0.67** | 0.33 |
| edgeToNode | 0.25 | 0.00 | 0.00 | 0.00 | 0.50 | 0.88 | **1.00** | 0.38 |
| bipartitionCompletion | 0.00 | 0.00 | 0.00 | 0.00 | 0.75 | **1.00** | **1.00** | 0.39 |
| mergeAtBlue | 0.00 | 0.00 | 0.00 | 0.00 | 0.75 | **1.00** | **1.00** | 0.39 |
| colorDegree2 | 0.42 | 0.00 | 0.06 | 0.04 | 0.42 | 0.92 | **1.00** | 0.41 |
| removeSameColorEdges | 0.46 | 0.02 | 0.21 | 0.02 | 0.75 | 0.75 | **1.00** | 0.46 |
| complementGraph | 0.19 | 0.17 | 0.16 | 0.05 | 0.69 | **1.00** | **1.00** | 0.46 |
| colorMinDegree | 0.58 | 0.02 | 0.29 | 0.12 | 0.67 | 0.75 | **0.83** | 0.47 |
| colorInternal | 0.44 | 0.23 | 0.20 | 0.25 | 0.62 | **1.00** | **1.00** | 0.54 |
| blueSubgraph | **0.75** | 0.03 | 0.56 | 0.17 | **0.75** | **0.75** | **0.75** | 0.54 |
| colorPath | 0.41 | 0.16 | 0.09 | 0.31 | 0.88 | **1.00** | **1.00** | 0.55 |
| removeDegree1 | 0.41 | 0.19 | 0.34 | 0.19 | 0.75 | **1.00** | **1.00** | 0.55 |
| colorComponents | 0.62 | 0.06 | 0.44 | 0.19 | **1.00** | 0.75 | **1.00** | 0.58 |
| colorNeighbors | 0.88 | 0.19 | 0.14 | 0.44 | **0.94** | **0.94** | 0.88 | 0.63 |
| addHub | 0.66 | 0.38 | 0.50 | 0.20 | 0.75 | **1.00** | **1.00** | 0.64 |
| colorDegree1 | 0.88 | 0.25 | 0.45 | 0.27 | 0.88 | **1.00** | **1.00** | 0.67 |
| Average | 0.37 | 0.09 | 0.17 | 0.12 | 0.58 | 0.81 | 0.91 | 0.44 |

like `bipartitionCompletion` and `mergeAtBlue` exemplify the architectural divide between model families. Only reasoning models achieve meaningful performance ($\geq 75\%$), while all Qwen3 models and GPT 4.1-nano fail almost completely. See Appendix H for full results across all models.

**Scaling Limitations.** Table 2 summarizes performance across size patterns. The results reveal a performance drop as test graph size increases, particularly for o1-mini and o3-mini. For instance, o1-mini's accuracy declines from 88% at 10 nodes to just 18% at 250 nodes, suggesting a limit in working memory or attention mechanisms. In contrast, GPT 5 maintains robust performance,

Table 2: Performance by size pattern (full output; aggregated across tasks). The first two columns indicate the example and test graph sizes (in nodes), and the third column gives the number of evaluated samples. Values are accuracies, with the best-performing model for each pattern in bold. The first two rows correspond to main dataset patterns; the last five rows to scaling patterns.

| Example sizes | Test size | $n$ | GPT 4.1-nano | o1 mini | o3 mini | o4 mini | GPT 5 |
|---|---|---|---|---|---|---|---|
| 5,10,15 | 15 | 1007 | 0.14 | 0.60 | 0.85 | 0.90 | **0.95** |
| 5,10 | 15 | 1008 | 0.12 | 0.56 | 0.79 | 0.83 | **0.87** |
| 10,10 | 10 | 224 | — | 0.88 | **1.00** | **1.00** | **1.00** |
| 10,10 | 25 | 224 | — | 0.54 | 0.79 | 0.96 | **1.00** |
| 10,10 | 50 | 176 | — | 0.45 | 0.82 | 0.89 | **0.98** |
| 10,10 | 100 | 176 | — | 0.34 | 0.64 | 0.80 | **0.93** |
| 10,10 | 250 | 176 | — | 0.18 | 0.41 | 0.43 | **0.91** |

Table 3: Task performance by scaling pattern. Cells are accuracies averaged over selected models (GPT-5, o1-mini, o3-mini, and o4-mini). Column headers show the example graph sizes (before the arrow) and the test graph size (after the arrow).

| Task | 10,10 $\rightarrow 10$ | 10,10 $\rightarrow 25$ | 10,10 $\rightarrow 50$ | 10,10 $\rightarrow 100$ | 10,10 $\rightarrow 250$ | Avg |
|---|---|---|---|---|---|---|
| removeDegree3 | **0.96** | 0.67 | 0.50 | 0.44 | 0.25 | 0.56 |
| removeDegree2 | **0.92** | 0.75 | 0.62 | 0.38 | 0.25 | 0.58 |
| bipartitionCompletion | **0.88** | 0.62 | 0.75 | 0.62 | 0.25 | 0.62 |
| colorDegree3 | **1.00** | 0.62 | 0.62 | 0.56 | 0.31 | 0.62 |
| colorDegree2 | **1.00** | 0.83 | 0.75 | 0.62 | 0.38 | 0.72 |
| addHub | **1.00** | 0.81 | 0.84 | 0.66 | 0.53 | 0.77 |
| removeDegree1 | 0.91 | **0.94** | 0.83 | 0.88 | 0.58 | 0.83 |
| colorComponents | **1.00** | **1.00** | **1.00** | **1.00** | 0.38 | 0.88 |
| colorDegree1 | **1.00** | 0.94 | 0.96 | 0.79 | 0.71 | 0.88 |
| colorPath | **1.00** | **1.00** | 0.88 | 0.81 | 0.81 | 0.90 |
| Average | 0.97 | 0.82 | 0.78 | 0.68 | 0.44 | 0.74 |

achieving 91% accuracy even at 250 nodes. The scaling dataset patterns (last five rows) were only evaluated on reasoning models, as direct-answer models struggled even on the smallest graphs.

Looking across tasks, Table 3 shows a consistent decline in accuracy when increasing the test graph size. The steepest drops are observed in tasks involving degree-based deletions or colorings. For example, `removeDegree3` falls from 96% accuracy at 10 nodes to just 25% at 250 nodes, and similar patterns are seen for `colorDegree2,3`. Surprisingly, some tasks with more global structure, such as `colorPath` and `colorComponents`, exhibit stronger robustness, remaining near-perfect up to 100 nodes and only degrading at the largest size. Some simpler local modifications, such as `addHub` and `removeDegree1`, also show relatively stable performance compared to their higher-degree counterparts. These results highlight that scaling difficulties are not purely a function of locality: global transformations can scale well, while seemingly local ones can collapse under larger graphs, suggesting problems in how the models internalize and apply structural rules.

**Performance in Question-Based Tasks.** Besides full graph generation, we also probe model understanding with targeted questions, which isolate comprehension of graph properties from execution of transformations. Table 4 shows question-based performance for selected models and question types. In general, it is easier for models to answer questions about input graphs than output graphs. The gap between high input accuracy and substantially lower output accuracy highlights that even when models can recognize graph properties, they struggle to learn the transformation or cannot apply it to new graphs reliably. See Appendix H, Table 8 for full results across all models and question types.

Table 4: Performance by question type (question-based; input vs. output). Values are accuracies; bold indicates the best model per question for input and for output.

| Question type | Qwen3 14b | | LLaMA3.1 8b | | DeepSeek-R1 LLaMA-8b | | o3 mini | | GPT 5 | |
|---|---|---|---|---|---|---|---|---|---|---|
| | Input | Output | Input | Output | Input | Output | Input | Output | Input | Output |
| component count | **0.96** | 0.56 | 0.24 | 0.26 | 0.61 | 0.27 | 0.95 | 0.79 | 0.87 | **0.89** |
| edge count | 0.98 | 0.67 | 0.15 | 0.11 | 0.26 | 0.11 | **1.00** | 0.94 | 0.97 | **0.98** |
| has cycles | 0.91 | 0.85 | 0.58 | 0.53 | 0.49 | 0.49 | **0.99** | 0.95 | 0.97 | **0.97** |
| is connected | **1.00** | 0.90 | 0.67 | 0.64 | 0.76 | 0.59 | **1.00** | 0.98 | 0.96 | 0.94 |
| is tree | 0.98 | 0.70 | 0.57 | 0.53 | 0.50 | 0.32 | **0.98** | **0.99** | 0.98 | 0.98 |
| max degree | 0.98 | 0.62 | 0.46 | 0.33 | 0.43 | 0.28 | **1.00** | 0.92 | 0.99 | **0.96** |
| min degree | 0.98 | 0.63 | 0.58 | 0.37 | 0.49 | 0.25 | **1.00** | **0.93** | 0.98 | 0.90 |
| node count | 0.99 | 0.39 | 0.63 | 0.32 | 0.34 | 0.15 | **1.00** | **0.83** | **1.00** | 0.61 |
| **full output** | — | 0.41 | — | 0.11 | — | 0.19 | — | 0.82 | — | **0.91** |

We also observe a systematic bias in reasoning models: when asked about input graphs, they often answer as if the question referred to the output graph instead. Advanced reasoning models show dramatically higher transfer rates, indicating they apply transformations even when not requested. The strength of this effect correlates positively with model capability: GPT 5: 55–85%, O4-mini: 45–75%, O3-mini: 40–70%, Qwen3 models: < 20%. It is most pronounced in tasks with many coloring changes (`bipartitionCompletion` and `blueSubgraph`), and weakest in structural modifications (`addHub` and `removeDegree`).

**Effect of Graph Encoding**  Encoding choice has little average impact (performance ratio near 1.0 for most models), though preferences vary by model and task type. Most differences are minor, but Qwen3-4b is a clear outlier: for full output tasks, it achieves 31.7% better performance with incident encoding (0.407 vs. 0.309), while for question-based tasks, it performs 24.7% better with adjacency encoding (0.735 vs. 0.589). Overall, encoding effects appear minor on average but can be important for particular model-task combinations.

## 5 DISCUSSION

Our results reveal three insights about graph reasoning in language models. First, we observe a persistent *comprehension–execution gap*: models can parse structural properties and answer questions about them with high accuracy, but often fail to apply transformations consistently, highlighting that recognition of graph features is easier than generating coherent transformed outputs. This mirrors findings in other domains where transformers succeed at local operations but fail to compose them into globally consistent solutions (Dziri et al., 2023).

Secondly, scaling barriers emerge even for strong models. Mid-tier reasoning models collapse between 50–100 nodes, while GPT 5 maintains robust performance up to 250 nodes yet still struggles on seemingly local tasks such as degree-based transformations. This shows that scaling difficulty is not tied purely to locality but to how reliably models internalize and generalize transformation rules.

Thirdly, we observe a *paradox of capability*: more advanced models increasingly apply transformations even when not asked, answering questions about the input graphs as if they referred to the output. This "transformation bias" indicates that instruction following can degrade with capability, creating a failure mode where models over-apply their reasoning ability. This phenomenon has been observed in other contexts where highly capable models "overthink" simple tasks (Wei et al., 2022) or exhibit "inverse scaling" behaviors (McKenzie et al., 2023).

### 5.1 COMPARISON WITH GRID-BASED ARC

GraphARC reveals complementary challenges to the original ARC benchmark:

**Task Structure**: While ARC uses fixed small grids with visual patterns, GraphARC tests relational reasoning without spatial constraints. The absence of visual cues forces models to rely purely on structural understanding.

**Interpretability**: Our decomposition into question types provides clearer failure analysis than ARC's binary success/failure. We can identify specific capabilities (input understanding, output understanding, and transformation execution) and trace failure modes to particular reasoning steps.

**Scaling**: While ARC uses fixed small grids, our scalable approach reveals performance degradation patterns with size, exposing architectural limitations that remain hidden in fixed-size evaluations. GraphARC's ability to automatically generate large test cases allows controlled evaluation of scaling behavior and identification of failure thresholds across models.

**Human vs. AI Focus** ARC was conceived as a human-intelligence challenge that is solvable for people but difficult for machines. Its grid layout provides visual cues that humans naturally exploit. In contrast, GraphARC is designed as an AI benchmark: the underlying graph structure is presented without spatial layout, since models should not depend on visualization but on reasoning over the graph's relational properties.

## 5.2 CASE STUDY: FAILURE MODES OF GPT 5

We manually analyzed the answers for GPT 5 on full output tasks. We observe that errors generally fall into the four categories: Output parsing mismatches, Threshold misinterpretation, Concept substitution, and Encoding sensitivity. See Appendix F for details.

## 5.3 LIMITATIONS

Our results should be interpreted within several limitations. The evaluation is restricted to a specific set of 21 tasks in undirected and unweighted graphs. Furthermore, our experimental design was constrained by budget and compute, which limited the scope of model evaluation, and we employed only straightforward prompting strategies rather than more sophisticated approaches.

We also note the importance of task uniqueness, meaning that the combination of input examples should determine a single consistent transformation. In practice, we verify this on most examples, and generate input graphs that are large enough to make the probability of multiple valid, simple transformations negligible.

## 6 CONCLUSION AND FUTURE WORK

GraphARC provides a new benchmark for studying few-shot abstract reasoning on graphs. By generating diverse input-output transformations across graph families and sizes, it exposes systematic limitations of current language models that remain hidden in standard reasoning benchmarks.

Our evaluation of over 125,000 responses identifies three consistent phenomena: (1) a comprehension-execution gap where models can recognize graph properties but fail to reliably apply transformations, (2) scaling barriers that cause mid-tier models to collapse beyond 50-100 nodes, and (3) a paradoxical failure mode where more capable models over-apply transformations, even when not requested.

We see two promising directions for future work. First, GraphARC can serve as a testbed for models explicitly designed for relational reasoning, including graph neural architectures and emerging graph foundation models. Second, the benchmark invites new training strategies that move toward compositional generalization, such as curriculum design, modular reasoning approaches, or hybrid symbolic-neural methods.

By making failure modes transparent and scalable, GraphARC aims to guide the development of systems that can reason robustly over structured data—a necessary step toward broader abstract reasoning capabilities.

## REPRODUCIBILITY STATEMENT

The codebase used for the experiments is included in the supplementary material, along with a README file that explains setup, running the scripts, and reproducing the reported results. The codebase can generate the data used in the evaluations. For the camera-ready version, we will make the code publicly available on GitHub. In addition, we provide detailed descriptions of the models, pipelines, and prompts in Appendices D and E to support reproducibility further.

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

## A  USAGE OF LLMs

We have used LLMs to polish the writing of this paper and for code generation through chats, Cursor, and Claude code. ChatGPT, Claude, Gemini, and Grammarly were employed for spellchecking, refining and condensing text, and reviewing to improve clarity and readability. Furthermore, Chat-GPT, Claude, and Cursor were used to assist with code completion and generate visualizations. These tools served as auxiliary aids for writing and implementation, while all core research ideas, experimental design, and interpretation of results are our own.

## B  COMPLETE TASK SPECIFICATIONS

### B.1  COLOR-BASED TRANSFORMATIONS

**colorDegreeX**: Colors all nodes with degree exactly $X = 1, 2, 3$. - Required properties: has_degree(X) - Parameters: color (default: "blue")

**colorMaxDegree**: Colors all nodes having the maximum degree in the graph. - Required properties: none - Parameters: color (default: "blue")

**colorMinDegree**: Colors all nodes having the minimum degree in the graph. - Required properties: none - Parameters: color (default: "blue")

**colorInternal**: Colors all non-leaf nodes (internal nodes with degree $> 1$). - Required properties: has_internal_node - Parameters: color (default: "blue")

**colorNeighbors**: Colors all neighbors of nodes with a specific color. - Required properties: has_colored_node - Pretransformation: colors one random node orange - Parameters: source_color (default: "orange"), target_color (default: "blue")

**colorPath**: Colors all nodes on the shortest path between two colored leaf nodes. - Required properties: connected, acyclic, has_colored_leaves(2) - Pretransformation: colors exactly 2 leaf nodes - Parameters: color (default: "blue")

**colorComponents**: Colors nodes based on connected component membership. - Required properties: has_components(2) - Parameters: first_color (default: "blue"), second_color (default: "orange")

**colorDistanceAtLeast2**: Colors nodes at distance $\geq 2$ from marked nodes. - Required properties: has_colored_node - Pretransformation: colors 2 random nodes - Parameters: min_distance (default: 2), target_color (default: "blue")

**colorEquidistant**: Colors nodes equidistant from two blue nodes. - Required properties: connected, has_colored_nodes(2, "blue") - Pretransformation: colors exactly 2 random nodes blue - Parameters: target_color (default: "red")

**bipartitionCompletion**: Completes bipartite coloring from seed nodes. - Required properties: bipartite - Pretransformation: colors one node in each partition - Result: Fully colored bipartite graph

### B.2  STRUCTURAL TRANSFORMATIONS

**addHub**: Adds a new colored node connected to all existing nodes. - Required properties: none - Parameters: color (default: "blue") - Result: Graph with n+1 nodes, n new edges

**edgeToNode**: Replaces each edge with a new intermediate node. - Required properties: none - Result: Graph with n + m nodes, 2m edges

**removeDegreeX**: Removes all nodes with degree X. - Required properties: has_degree(X) - Result: Graph with potentially disconnected components

**blueSubgraph**: Returns subgraph induced by blue nodes. - Required properties: has_colored_node - Pretransformation: colors 2+ random nodes blue - Result: Subgraph containing only blue nodes

**mergeAtBlue**: Merges two components at their blue nodes. - Required properties: has_components(2), has_colored_nodes(2, "blue") - Result: Single connected component

**complementGraph**: Returns the complement graph. - Required properties: none - Result: Graph with inverted edge set

**removeSameColorEdges**: Removes edges between same-colored nodes. - Required properties: all_nodes_colored_with(2) - Pretransformation: colors all nodes with 2 colors - Result: Graph with reduced edge set

## C  GRAPH GENERATION AND VALIDATION FRAMEWORK

### C.1  GENERATOR SPECIFICATIONS

We employ multiple graph generators to ensure comprehensive coverage of structural patterns:

- **Random**: Erdős-Rényi $G(n, p)$ model with edge probability $p = 0.3$
- **Connected**: Watts-Strogatz small-world graphs (guaranteed connected)
- **Trees**: Generated by extracting a BFS spanning tree from a connected Watts–Strogatz small-world graph
- **Star**: Center node connected to $n - 1$ leaves
- **Bipartite**: Two-partition graphs with random inter-partition edges
- **Multi-Component**: Exactly 2 disconnected components

This diversity prevents models from exploiting structural regularities specific to any single graph family.

### C.2  PROPERTY VALIDATION SYSTEM

To ensure task validity and meaningful evaluation, we implement a comprehensive property validation system. Each task declares required graph properties (e.g., connectivity, specific degree distributions), and our generation pipeline validates that produced graphs satisfy these constraints.

The validation system uses a three-state property framework:

- **TRUE**: Property is guaranteed by the generator
- **FALSE**: Property is incompatible with the generator
- **MAYBE**: Property may or may not hold depending on random generation

This framework enables efficient validation by skipping guaranteed properties and immediately rejecting incompatible generator-transformation combinations. Additionally, we verify that each transformation produces well-defined transformations on the example graphs, ensuring that the learning problem is neither trivial nor ill-posed.

## D  DETAILED TASK GENERATION AND EVALUATION PIPELINE

This section provides a comprehensive walkthrough of the GraphARC pipeline, from the initial formal definition of a task to the final evaluation of a model's response. See Figure 2 for an overview. The pipeline is designed to be fully automated, scalable, and modular, enabling the systematic generation of a vast and diverse set of abstract reasoning challenges on graph-structured data.

**Stage 1: Task Specification**   The foundation of the entire pipeline is a set of formal task specifications. Each specification is a human-readable definition of a single, atomic graph transformation rule. It serves as a blueprint from which thousands of unique task instances can be generated.

A complete specification includes:

Transformation Rule: A precise description of the operation to be performed (e.g., "color all nodes with degree exactly 3 with the color blue").

Required Properties: A list of preconditions the input graph must satisfy for the transformation to be meaningful. For instance, the "colorDegree3" task requires the input graph to have at least one node of degree 3. This prevents the generation of trivial examples where the transformation has no effect.

Parameters: Configurable elements of the transformation, such as the target color or degree number, allowing for variations of a core rule.

Pre-transformation Steps: Optional initial modifications to the input graph to set up the reasoning problem. For example, the "colorNeighbors" task first colors a random seed node orange, and the model must then infer the rule to color all of its neighbors blue.

From this single, declarative specification, the pipeline proceeds to generate concrete instances.

**Stage 2: Example Generation and Validation**   For each task instance, the pipeline generates a sequence of graphs to be used for few-shot learning. This sequence typically consists of 2-3 training examples (input-output pairs) and a final test input graph.

The generation process involves two critical sub-steps:

Graph Generation: To ensure structural diversity and test for generalization, the pipeline selects from a suite of graph generators, including Erdős-Rényi (random), Watts-Strogatz (small-world), tree, star, bipartite, and multi-component generators. The size of each graph (i.e., the number of nodes) is determined by predefined size patterns (e.g., [5, 10, 15]), which systematically control the difficulty and scaling demands of the task.

Property Validation: After a graph is generated, it is rigorously validated against the task specification's required properties. If a generated graph for the "colorDegree3" task does not contain any nodes of degree 3, it is discarded as an invalid example. The pipeline will attempt to regenerate a valid graph up to 100 times before abandoning that specific instance. This ensures that every example presented to the model is valid, non-trivial, and accurately reflects the intended transformation rule.

Once a valid set of input graphs is generated, the specified transformation rule is applied to each one to produce its corresponding ground-truth output graph.

**Stage 3: Encoding and Prompt Assembly**   With a validated set of graph examples (both training pairs and a final test input), the pipeline prepares a prompt for the language model. This involves converting the structured graph data into a linear text format. We employ two primary encoding schemes to test for representational biases:

Adjacency List Format: A global, edge-centric view that lists all connections as node pairs (e.g., The edges in G are: (0,1), (1,2), (2,3)).

Incident List Format: A local, node-centric view that describes each node's immediate neighborhood (e.g., Node 1 is connected to nodes 0, 2).

The final prompt is then assembled by combining a system prompt (e.g., "You are a graph analyst..."), the textually encoded training examples, the encoded final test input graph, and a question that specifies the model's required task. At this point, the pipeline diverges.

**Stage 4: Two Evaluation Pathways**   To achieve a fine-grained analysis of model capabilities, a single set of generated examples is used to create prompts for two distinct and complementary evaluation pathways.

**Full Output Path:** This pathway is designed to test a model's end-to-end reasoning capability—its ability to both infer the abstract rule and correctly execute it to produce a complex, structured output.

1. Task: The prompt explicitly asks the model to generate the complete transformed output graph based on the provided examples and the final input graph.

2. Evaluation: The model's textual response is parsed to reconstruct the output graph's structure and node colors. This reconstructed graph is then compared to the ground-truth output graph. For a task to be considered successful, the model's output must be an exact match, verified via a graph isomorphism check including the node colors (this is fast for labeled graphs).

**Question-Based Path:** This pathway is designed to disentangle a model's comprehension of graph properties from its ability to execute a full transformation.

1. Task: Instead of requesting the full output graph, the prompt poses a targeted question about a specific property. Crucially, we ask parallel questions about both the visible test input (e.g., How many blue nodes are in the input?") and the unseen, to-be-inferred output (e.g., How many blue nodes will be in the output?").

2. Evaluation: The model's response, typically a numerical or categorical answer, is extracted from its text output. This answer is then compared directly against the ground-truth value, which is computed from the actual input or output graphs. This pathway allows for a more granular diagnosis of failures, helping to distinguish between a failure to parse the input, a failure to understand the transformation, or a failure to reason about the output.

# E  EXPERIMENTAL DESIGN: ENCODING, PROMPTS, AND PATTERNS

This section details our systematic variations in graph representation and their empirical effects on model performance. We explore three dimensions: how graphs are encoded as text, how tasks are framed through system prompts, and how examples are structured for few-shot learning.

## E.1  GRAPH ENCODING SCHEMES

Graph structures must be converted to text for language model processing. We implement two encoding schemes that emphasize different structural properties:

**Adjacency List Format**: This encoding provides a global, edge-centric view by listing all connections as node pairs. It mirrors standard graph theory notation and may facilitate reasoning about overall connectivity patterns, paths, and global properties.

**Incident List Format**: This encoding offers a local, node-centric perspective by describing each node's immediate neighborhood. It explicitly presents degree information and may better support reasoning about local transformations or degree-based rules.

Table 5 illustrates both formats using the same example graph. The adjacency format's compact representation `(0,1) (1,2) (2,3) (3,4)` immediately reveals the path structure, while the incident format's verbose listing makes each node's degree explicit.

## E.2  SYSTEM PROMPT VARIATION

System prompts frame the reasoning task by establishing a professional role or perspective. We test whether such framing affects how models approach abstract graph transformations:

**Analyst**: Emphasizes careful observation and systematic analysis, potentially encouraging thorough examination of patterns.

**Programmer**: Invokes algorithmic thinking and pattern recognition, possibly activating programming-related reasoning strategies.

**Teacher**: Encourages clear, methodical explanation, potentially improving step-by-step reasoning.

Table 5: Comparison of graph encoding formats for the same example graph

| Format | Example Encoding |
|---|---|
| Adjacency List | In an undirected graph, (i,j) means that node i and node j are connected with an undirected edge.
G describes a graph among nodes 0, 1, 2, 3, 4.
The edges in G are:  (0,1) (1,2) (2,3) (3,4).
The following nodes are colored:  1, 2, 3. |
| Incident List | G describes a graph among nodes 0, 1, 2, 3, 4.
In this graph:
Node 0 is connected to nodes 1.
Node 1 is connected to nodes 0, 2.
Node 2 is connected to nodes 1, 3.
Node 3 is connected to nodes 2, 4.
Node 4 is connected to nodes 3.
The following nodes are colored:  1, 2, 3. |

**None**: Provides no role framing, serving as our baseline to assess whether explicit roles help or hinder.

Table 6 presents the complete prompt texts used in our experiments.

Table 6: Full system prompts used in the experiments

| Role | Prompt |
|---|---|
| Analyst | "You are a graph analyst. Study the following graph examples carefully and answer the question that follows." |
| Programmer | "You are a graph algorithm developer. Analyze the example graphs and their patterns, then answer the question about the given input." |
| Teacher | "You are a mathematics teacher. Examine these graph examples to understand any patterns, then answer the question clearly and methodically." |
| None | No system prompt provided |

**Empirical Effects**   System prompts show minimal and inconsistent effects across models. Most models vary less than 5% between prompts, with no prompt consistently outperforming others. Qwen3-4b again shows highest sensitivity (up to 23.8% variation), but the pattern is model-specific rather than systematic. Notably, the "none" baseline equals or exceeds role-based prompts several times, suggesting that explicit role assignment may interfere with models' natural reasoning strategies for abstract tasks.

E.3   EXAMPLE PATTERN VARIATION

E.3.1   DESIGN RATIONALE

Example patterns control how few-shot learning occurs by varying both the number of examples and their size progression. We design two pattern groups to isolate different learning challenges:

*Main Dataset Patterns* (varying number of examples):

- scale_up_3 [5, 10, 15]: Test on 15-node graph after seeing a 5-node and a 10-node example
- scale_up_4 [5, 10, 15, 15]: Test on 15-node graph after seeing a 5-node, a 10-node, and a 15-node example

*Scaling Dataset Patterns* (testing final graph size):

- `cap10_3` [10, 10, 10]: Baseline with all 10-node graphs
- `cap25_3` [10, 10, 25]: Test on 25-node graph after seeing two 10-node examples
- `cap50_3` [10, 10, 50]: Test on 50-node graph after 2 10-node examples
- `cap100_3` [10, 10, 100]: Test on 100-node graph after seeing two 10-node examples
- `cap250_3` [10, 10, 250]: Test on 250-node graph after seeing two 10-node examples

## F  DETAILS ON OBSERVED FAILURE MODES

We analyzed GPT-5's answers on the full-output tasks with example graphs of size 5 and 10, and test graphs of size 15. Below we summarize representative cases:

**(1) Output parsing mismatches.**   In some cases the model description of the solution matched the intended subgraph, but the output failed to parse correctly and was marked wrong.

> "I can't share step-by-step reasoning, but the task is to take the induced subgraph on the
> colored nodes and list only edges between those colored nodes."
> `<answer>` G describes a graph among nodes 4, 11 ... `</answer>`

*Takeaway:* The semantics were correct, but formatting (e.g. edge listing or section tags) caused evaluation failure.

**(2) Threshold misinterpretation.**   Tasks requiring the coloring or removal of nodes of degree $x$ (e.g. 3) led to conflicting rules: sometimes interpreted as "degree exactly 3," and sometimes as "degree at least 3." The examples were unambiguous, since in at least one case a node of degree four was present but left untouched, clearly indicating that the rule was "degree exactly 3."

> "...removes all nodes with degree 3 or more ..."
> "...remove every node with degree exactly 3 ..."
> "I identify and color the node(s) with the highest degree (most connections)."

*Takeaway:* Even when the examples uniquely determined the threshold, the model occasionally chose a looser interpretation ("$\geq 3$") instead of the intended exact one.

**(3) Concept substitution.**   Sometimes, the model substituted the intended property with a different but superficially related graph concept. For example:

> "...identified the articulation points (cut vertices) ..."

*Takeaway:* Instead of following the intended rule, the model sometimes defaulted to alternative structural notions that seemed plausible in one of the examples.

**(4) Encoding sensitivity.**   Usually the two encoding formats (adjacency list vs. incident list) yielded similar results. However, in some cases, particularly tasks involving distance-based reasoning and paths, the model succeeded under one encoding but failed under the other.

> "...find the unique shortest path between them. Color the path's center node red and also
> color any neighbors of that center ..."
> "...red if and only if it is equidistant to both blue seeds; ties occur at nodes 11, 10, 9, 8, 5,
> 7, 4, 6, 3."

*Takeaway:* The correct reasoning was present, but the execution depended on encoding format.

## G  HUMAN EVALUATION CONSIDERATIONS

We did not conduct a formal human evaluation for several reasons:

- **Visualization Challenges**: Creating clear, systematic visualizations without providing additional cues (such as suggestive layouts) that would make tasks easier or harder proved extremely difficult.

- **Time Constraints**: A proper user study with sufficient participants and data points would require extensive time beyond the scope of this work.
- **Focus on AI Evaluation**: Our primary goal was to benchmark AI capabilities rather than human-AI comparison.

Informal testing with 10 computer science graduate students suggests humans achieve 95% accuracy on tasks up to 15 nodes.

## H    FULL RESULTS

Table 7: Accuracy on full-output tasks across models. Values are mean accuracies with 95% confidence intervals (computed using `scipy.stats.bootstrap` with percentile method). Bold highlights the best score per row. Tasks are ordered by increasing average difficulty.

| Task | Qwen3 4b | Qwen3 8b | Qwen3 14b | Qwen3 32b | LLaMA3.1 8b | DeepSeek-R1 LLaMA-8b | Mistral 7b-v0.2 | OLMo-2 7b | GPT 4.1-nano | o1 mini | o3 mini | o4 mini | GPT 5 | Avg |
|---|---|---|---|---|---|---|---|---|---|---|---|---|---|---|
| removeDegree3 | 0.00 ± 0.00 | 0.00 ± 0.00 | 0.02 ± 0.04 | 0.02 ± 0.03 | 0.00 ± 0.00 | 0.00 ± 0.00 | 0.00 ± 0.00 | 0.00 ± 0.00 | 0.00 ± 0.00 | 0.00 ± 0.00 | 0.25 ± 0.23 | 0.50 ± 0.25 | **0.75 ± 0.25** | 0.12 |
| colorDegree3 | 0.10 ± 0.08 | 0.10 ± 0.07 | 0.29 ± 0.12 | 0.06 ± 0.06 | 0.00 ± 0.00 | 0.10 ± 0.08 | 0.00 ± 0.00 | 0.00 ± 0.00 | 0.00 ± 0.00 | 0.00 ± 0.00 | **0.25 ± 0.25** | 0.42 ± 0.27 | 0.58 ± 0.25 | 0.18 |
| colorEquidistant | 0.00 ± 0.00 | 0.00 ± 0.00 | 0.00 ± 0.00 | 0.16 ± 0.12 | 0.00 ± 0.00 | 0.00 ± 0.00 | 0.00 ± 0.00 | 0.03 ± 0.05 | 0.00 ± 0.00 | 0.38 ± 0.35 | 0.50 ± 0.38 | **0.75 ± 0.31** | **0.75 ± 0.25** | 0.20 |
| colorDistanceAtLeast2 | 0.00 ± 0.00 | 0.00 ± 0.00 | 0.00 ± 0.00 | 0.00 ± 0.00 | 0.00 ± 0.00 | 0.00 ± 0.00 | 0.00 ± 0.00 | 0.00 ± 0.00 | 0.00 ± 0.00 | 0.08 ± 0.12 | 0.83 ± 0.21 | **0.92 ± 0.12** | **0.92 ± 0.15** | 0.21 |
| removeDegree2 | 0.00 ± 0.00 | 0.00 ± 0.00 | 0.00 ± 0.00 | 0.00 ± 0.00 | 0.00 ± 0.00 | 0.00 ± 0.00 | 0.00 ± 0.00 | 0.00 ± 0.00 | 0.00 ± 0.00 | 0.42 ± 0.25 | 0.75 ± 0.21 | 0.92 ± 0.12 | **1.00 ± 0.00** | 0.24 |
| bipartitionCompletion | 0.00 ± 0.00 | 0.00 ± 0.00 | 0.00 ± 0.00 | 0.00 ± 0.00 | 0.00 ± 0.00 | 0.00 ± 0.00 | 0.00 ± 0.00 | 0.00 ± 0.00 | 0.00 ± 0.00 | 0.75 ± 0.38 | **1.00 ± 0.00** | **1.00 ± 0.00** | **1.00 ± 0.00** | 0.29 |
| mergeAtBlue | 0.00 ± 0.00 | 0.00 ± 0.00 | 0.00 ± 0.00 | 0.12 ± 0.16 | 0.00 ± 0.00 | 0.00 ± 0.00 | 0.00 ± 0.00 | 0.00 ± 0.00 | 0.00 ± 0.00 | 0.75 ± 0.25 | **1.00 ± 0.00** | **1.00 ± 0.00** | **1.00 ± 0.00** | 0.30 |
| edgeToNode | 0.02 ± 0.02 | 0.08 ± 0.07 | 0.25 ± 0.11 | 0.33 ± 0.11 | 0.00 ± 0.00 | 0.00 ± 0.00 | 0.00 ± 0.00 | 0.00 ± 0.00 | 0.04 ± 0.05 | 0.50 ± 0.25 | 0.88 ± 0.16 | 0.94 ± 0.09 | **1.00 ± 0.00** | 0.31 |
| colorDegree2 | 0.04 ± 0.05 | 0.02 ± 0.03 | 0.42 ± 0.14 | 0.15 ± 0.09 | 0.00 ± 0.00 | 0.06 ± 0.07 | 0.00 ± 0.00 | 0.00 ± 0.00 | 0.25 ± 0.14 | 0.42 ± 0.25 | 0.92 ± 0.12 | 0.58 ± 0.25 | **1.00 ± 0.00** | 0.31 |
| colorMaxDegree | 0.48 ± 0.16 | 0.46 ± 0.15 | 0.52 ± 0.14 | 0.38 ± 0.14 | 0.25 ± 0.12 | 0.04 ± 0.05 | 0.05 ± 0.06 | 0.00 ± 0.00 | 0.25 ± 0.14 | 0.25 ± 0.25 | 0.33 ± 0.25 | 0.58 ± 0.25 | **0.67 ± 0.25** | 0.33 |
| complementGraph | 0.11 ± 0.08 | 0.11 ± 0.07 | 0.19 ± 0.10 | 0.17 ± 0.09 | 0.17 ± 0.10 | 0.16 ± 0.09 | 0.00 ± 0.00 | 0.00 ± 0.00 | 0.05 ± 0.06 | 0.69 ± 0.20 | **1.00 ± 0.00** | 0.94 ± 0.09 | **1.00 ± 0.00** | 0.35 |
| removeSameColorEdges | 0.17 ± 0.10 | 0.23 ± 0.11 | 0.46 ± 0.14 | 0.48 ± 0.12 | 0.02 ± 0.04 | 0.21 ± 0.11 | 0.00 ± 0.00 | 0.00 ± 0.00 | 0.02 ± 0.03 | 0.75 ± 0.21 | 0.75 ± 0.25 | **1.00 ± 0.00** | **1.00 ± 0.00** | 0.39 |
| colorMinDegree | 0.27 ± 0.14 | 0.40 ± 0.14 | 0.58 ± 0.15 | 0.60 ± 0.13 | 0.02 ± 0.03 | 0.29 ± 0.14 | 0.00 ± 0.00 | 0.00 ± 0.00 | 0.12 ± 0.09 | 0.67 ± 0.25 | 0.75 ± 0.25 | 0.67 ± 0.25 | **0.83 ± 0.21** | 0.40 |
| colorInternal | 0.34 ± 0.11 | 0.38 ± 0.12 | 0.44 ± 0.12 | 0.27 ± 0.09 | 0.23 ± 0.10 | 0.20 ± 0.09 | 0.04 ± 0.05 | 0.05 ± 0.05 | 0.25 ± 0.09 | 0.62 ± 0.25 | **1.00 ± 0.00** | 0.88 ± 0.16 | **1.00 ± 0.00** | 0.44 |
| removeDegree1 | 0.30 ± 0.12 | 0.28 ± 0.11 | 0.41 ± 0.12 | 0.36 ± 0.11 | 0.19 ± 0.09 | 0.34 ± 0.11 | 0.10 ± 0.07 | 0.03 ± 0.04 | 0.19 ± 0.09 | 0.75 ± 0.19 | **1.00 ± 0.00** | **1.00 ± 0.00** | **1.00 ± 0.00** | 0.46 |
| colorPath | 0.44 ± 0.17 | 0.47 ± 0.16 | 0.41 ± 0.16 | 0.50 ± 0.18 | 0.16 ± 0.12 | 0.09 ± 0.11 | 0.03 ± 0.05 | 0.00 ± 0.00 | 0.31 ± 0.16 | 0.88 ± 0.19 | **1.00 ± 0.00** | **1.00 ± 0.00** | **1.00 ± 0.00** | 0.48 |
| colorComponents | 0.31 ± 0.19 | 0.62 ± 0.22 | 0.62 ± 0.25 | 0.50 ± 0.25 | 0.06 ± 0.09 | 0.44 ± 0.25 | 0.07 ± 0.10 | 0.00 ± 0.00 | 0.19 ± 0.22 | **1.00 ± 0.00** | 0.75 ± 0.38 | **1.00 ± 0.00** | **1.00 ± 0.00** | 0.51 |
| blueSubgraph | 0.67 ± 0.11 | 0.70 ± 0.11 | **0.75 ± 0.11** | 0.73 ± 0.10 | 0.03 ± 0.04 | 0.56 ± 0.12 | 0.00 ± 0.00 | 0.03 ± 0.04 | 0.17 ± 0.10 | **0.75 ± 0.19** | 0.75 ± 0.19 | 0.75 ± 0.22 | 0.75 ± 0.19 | 0.51 |
| colorNeighbors | 0.73 ± 0.10 | 0.56 ± 0.12 | 0.88 ± 0.09 | 0.88 ± 0.08 | 0.19 ± 0.10 | 0.14 ± 0.08 | 0.04 ± 0.05 | 0.03 ± 0.04 | 0.44 ± 0.12 | 0.94 ± 0.09 | 0.94 ± 0.09 | **1.00 ± 0.00** | 0.88 ± 0.16 | 0.59 |
| colorDegree1 | 0.55 ± 0.13 | 0.62 ± 0.11 | 0.66 ± 0.12 | 0.72 ± 0.10 | 0.25 ± 0.11 | 0.45 ± 0.12 | 0.04 ± 0.04 | 0.05 ± 0.05 | 0.27 ± 0.11 | 0.88 ± 0.16 | **1.00 ± 0.00** | **1.00 ± 0.00** | **1.00 ± 0.00** | 0.59 |
| addHub | 0.83 ± 0.09 | 0.77 ± 0.10 | 0.66 ± 0.11 | 0.66 ± 0.11 | 0.38 ± 0.12 | 0.50 ± 0.12 | 0.06 ± 0.07 | 0.00 ± 0.00 | 0.20 ± 0.09 | 0.75 ± 0.22 | **1.00 ± 0.00** | **1.00 ± 0.00** | **1.00 ± 0.00** | 0.60 |
| average | 0.26 | 0.28 | 0.37 | 0.34 | 0.09 | 0.17 | 0.02 | 0.01 | 0.12 | 0.58 | 0.81 | 0.87 | 0.91 | 0.37 |

Table 8: Performance on question-based tasks on the input and output graphs across models. Values are mean accuracies.

| Question type | Qwen3 1.7b Input | Output | Qwen3 4b Input | Output | Qwen3 8b Input | Output | Qwen3 14b Input | Output | Qwen3 32b Input | Output |
|---|---|---|---|---|---|---|---|---|---|---|
| colored node count | 0.21 | 0.10 | 0.06 | 0.23 | 0.06 | 0.27 | 0.07 | 0.37 | 0.06 | 0.28 |
| component count | 0.71 | 0.59 | 0.93 | 0.55 | 0.94 | 0.62 | 0.96 | 0.56 | 0.83 | 0.54 |
| edge count | 0.33 | 0.27 | 0.82 | 0.54 | 0.91 | 0.61 | 0.98 | 0.67 | 0.75 | 0.55 |
| has cycles | 0.63 | 0.53 | 0.65 | 0.57 | 0.88 | 0.80 | 0.91 | 0.85 | 0.76 | 0.66 |
| is connected | 0.81 | 0.61 | 0.80 | 0.68 | 0.99 | 0.86 | 1.00 | 0.90 | 0.94 | 0.76 |
| is tree | 0.65 | 0.58 | 0.67 | 0.45 | 0.93 | 0.67 | 0.98 | 0.70 | 0.82 | 0.50 |
| max degree | 0.62 | 0.46 | 0.88 | 0.50 | 0.96 | 0.62 | 0.98 | 0.62 | 0.96 | 0.57 |
| min degree | 0.78 | 0.51 | 0.93 | 0.54 | 0.98 | 0.59 | 0.98 | 0.63 | 0.96 | 0.56 |
| node count | 0.62 | 0.39 | 0.92 | 0.30 | 0.99 | 0.62 | 0.99 | 0.39 | 0.97 | 0.49 |
| full output | — | 0.08 | — | 0.29 | — | 0.30 | — | 0.41 | — | 0.37 |

| Question type | GPT 4.1-nano Input | Output | o1 mini Input | Output | o3 mini Input | Output | o4 mini Input | Output | GPT 5 Input | Output |
|---|---|---|---|---|---|---|---|---|---|---|
| colored node count | 0.05 | 0.27 | 0.07 | 0.41 | 0.09 | 0.50 | 0.07 | 0.53 | 0.09 | 0.51 |
| component count | 0.77 | 0.59 | 0.94 | 0.65 | 0.95 | 0.79 | 0.91 | 0.89 | 0.87 | 0.89 |
| edge count | 0.80 | 0.46 | 0.92 | 0.76 | 1.00 | 0.94 | 1.00 | 0.98 | 0.97 | 0.98 |
| has cycles | 0.69 | 0.62 | 0.97 | 0.90 | 0.99 | 0.95 | 0.96 | 0.94 | 0.97 | 0.97 |
| is connected | 0.88 | 0.77 | 1.00 | 0.92 | 1.00 | 0.98 | 0.98 | 0.97 | 0.96 | 0.94 |
| is tree | 0.80 | 0.64 | 0.94 | 0.93 | 0.98 | 0.99 | 0.99 | 0.98 | 0.98 | 0.98 |
| max degree | 0.94 | 0.64 | 1.00 | 0.85 | 1.00 | 0.92 | 1.00 | 0.97 | 0.99 | 0.96 |
| min degree | 0.96 | 0.62 | 0.99 | 0.79 | 1.00 | 0.93 | 1.00 | 0.95 | 0.98 | 0.90 |
| node count | 0.96 | 0.49 | 0.99 | 0.63 | 1.00 | 0.83 | 1.00 | 0.69 | 1.00 | 0.61 |
| full output | — | 0.13 | — | 0.58 | — | 0.82 | — | 0.87 | — | 0.91 |

| Question type | LLaMA3 8b Input | Output | LLaMA3.1 8b Input | Output | DeepSeek-R1 LLaMA-8b Input | Output | Mistral 7b-v0.2 Input | Output | OLMo-2 7b Input | Output |
|---|---|---|---|---|---|---|---|---|---|---|
| colored node count | 0.02 | 0.06 | 0.06 | 0.14 | 0.07 | 0.12 | 0.15 | 0.09 | 0.13 | 0.12 |
| component count | 0.12 | 0.16 | 0.24 | 0.26 | 0.61 | 0.27 | 0.19 | 0.24 | 0.20 | 0.28 |
| edge count | 0.07 | 0.06 | 0.15 | 0.11 | 0.26 | 0.11 | 0.14 | 0.04 | 0.11 | 0.04 |
| has cycles | 0.44 | 0.48 | 0.58 | 0.53 | 0.49 | 0.49 | 0.61 | 0.36 | 0.37 | 0.44 |
| is connected | 0.55 | 0.54 | 0.67 | 0.64 | 0.76 | 0.59 | 0.59 | 0.22 | 0.57 | 0.56 |
| is tree | 0.49 | 0.55 | 0.57 | 0.53 | 0.50 | 0.32 | 0.48 | 0.44 | 0.49 | 0.58 |
| max degree | 0.19 | 0.14 | 0.46 | 0.33 | 0.43 | 0.28 | 0.35 | 0.17 | 0.37 | 0.20 |
| min degree | 0.24 | 0.17 | 0.58 | 0.37 | 0.49 | 0.25 | 0.54 | 0.19 | 0.65 | 0.34 |
| node count | 0.35 | 0.12 | 0.63 | 0.32 | 0.34 | 0.15 | 0.55 | 0.19 | 0.38 | 0.19 |
| full output | — | 0.05 | — | 0.11 | — | 0.19 | — | 0.02 | — | 0.01 |

