# OpenReview forum: "GraphARC: A Comprehensive Benchmark for Graph-Based Abstract Reasoning"
_ICLR.cc/2026/Conference — ICLR 2026 Conference Withdrawn Submission_

### Official Review · Reviewer_j5j2 · 2025-10-31

**Soundness:** 2
**Presentation:** 3
**Contribution:** 2
**Rating:** 4
**Confidence:** 3

**Summary:**

The paper introduces a graph-structured analogue of ARC. Each task gives 2-3 input-output graph pairs that exemplify a transformation rule, and the model must apply the rule to a new test graph. The benchmark spans 21 transformations over diverse graph families and sizes, and evaluates both end-to-end “full-output” generation and targeted “question-based” probes to separate comprehension from execution. Across SOTA LLMs, the authors report a comprehension-execution gap (models answer property questions much more easily than they synthesize the full transformed graph) and a consistent performance drop as test graphs scale. They also observe an input–output asymmetry (questions about the visible input > questions about the inferred output) and a transformation bias.

**Strengths:**

1. Precise task taxonomy (21 transformations) with formal properties and generators across multiple graph families supports systematic variation.

2. Task-specific performance analysis pinpoints which rules are robust and which remain brittle, informing method development.

3. Failure-mode inspection provides guidance for improving prompting and parsers.

**Weaknesses:**

1. No non-LLM baselines like GNNs/neural algorithmic reasoning methods. To ground difficulty and isolate modality effects, the benchmark should compare to strong graph-native methods, e.g., transformer-based neural algorithmic reasoning on graph algorithms or standard GNNs (GCN/GIN/Graphormer) fine-tuned to output transformations. This is necessary because such baselines test whether failures are LLM-specific or fundamental to the task. Also, since the paper positions GraphARC as a testbed for GFMs, include at least one graph-text model (e.g., GIMLET) or a simple GNN+LLM tool-use baseline (LLM plans, GNN executes). This is necessary to verify whether graph-native modules close the execution gap.

2. The paper should situate comparison vs related-benchmarks like NLGraph and GraphArena. Even if task formats differ, reporting head-to-head LLM results on their suites would clarify what GraphARC adds beyond existing graph-reasoning benchmarks.

3. The paper states uniqueness is “verified on most examples,” but does not quantify residual ambiguity rates or show adversarial counterexamples. Please report an explicit uniqueness score per task and regenerate until uniqueness ≥ 99.5%; otherwise, failures may reflect underdetermined rules.

4. Undirected/unweighted graphs only limits external validity (many applications depend on directions/weights/labels). Include a small directed-/weighted-graph extension or at least show a subset of tasks (e.g., shortest-path coloring) with edge directions to test rule-following under asymmetry would be useful.

5. Scaling claims would benefit from compute/memory controls. For the size-scaling study, normalize token budgets and show accuracy vs. effective context utilization; otherwise, performance drops might largely reflect context overflow, not reasoning per se.

**Questions:**

Will the paper report human accuracy/time on small-n tasks to contextualize sample efficiency?

---

### Official Review · Reviewer_ADS1 · 2025-11-01

**Soundness:** 3
**Presentation:** 3
**Contribution:** 1
**Rating:** 2
**Confidence:** 4

**Summary:**

The paper introduces GraphARC, a new benchmark for evaluating graph-based abstract reasoning in large language models.
It provides few-shot tasks where models must infer transformation rules from example input/output graph pairs and apply them to new graphs.

**Strengths:**

- Clear problem framing: few-shot graph transformations inspired by ARC; tasks span local, global, and hierarchical rules.

- Evaluates both full transformation and targeted Q&A, separating “execution” from “comprehension.”

- Reports a useful comprehension–execution gap and scaling failures; actionable signals for future work.

**Weaknesses:**

- Novelty is thin: largely ports ARC’s few-shot paradigm to graphs; similar prior lines already test LLMs on textual graph reasoning/pattern ops.

- Limited graph scope (undirected, simple transformations): misses richer settings (typed/weighted/directed edges, constraints, compositions).

- Benchmark centers LLM prompting: no head-to-head with graph-native baselines (GNNs / emerging GFMs) to calibrate difficulty and no advantages/disadvantages of incorporating GNN with LLMs.

- Representation choices (adjacency/incident lists) are known confounders: paper doesn’t deeply tackle index/ordering invariance beyond light encoding tests.

- Evaluation may overfit to small toy rules. Unclear transfer to realistic graph tasks (typed relations, hierarchical programs, multi-step composition).

**Questions:**

-  How is GraphARC fundamentally different from earlier textual graph-reasoning benchmarks that already test detection/translation/modification of patterns—beyond adopting the ARC few-shot interface? Please delineate concrete new capabilities or phenomena only GraphARC reveals.
- Did you test node-ID permutations and input ordering shuffles? If so, what’s the sensitivity breakdown per task type; if not, can you add it to isolate structural vs token-order cues?
- Why no GNN/GFM baselines? Many papers already acknowledged the limited reasoning of LLMs on graph tasks and worked to improve the performance of LLMs on these tasks.
- Any variants with typed/directed/weighted edges or heterogeneous nodes? If the goal is a stepping stone to GFMs, what’s the roadmap to those richer settings?

---

### Official Review · Reviewer_XWE3 · 2025-11-01

**Soundness:** 2
**Presentation:** 3
**Contribution:** 2
**Rating:** 4
**Confidence:** 3

**Summary:**

The paper proposes GraphARC benchmark for few-shot abstract reasoning on graphs. By generating tasks across multiple graph families and sizes and forcing models to infer rules from 2 to 3 graph input-output pairs, the benchmark enables studying whether models can truly apply a structural rule to a new graph, not just recognize local properties. The experiments on current LLMs show that models answer property questions far better than they can produce a full transformed graph, and the performance drops sharply with graph size for mid-tier reasoning models.

**Strengths:**

1. The paper is well-written.
2. The benchmark structure is well-defined.
2. The evaluation is comprehensive, including diverse transformations and, scaling study.

**Weaknesses:**

1. I don't feel this direction is very promising. The results look like the LLMs are just out of their distributions. Without fine-tuning, the comparisons are not meaningful enough.
2. A human study can make the paper stronger.
3. As a benchmark, there should be discussions about how to keep leaderboards and how to avoid data leaking, given the current LLM competition situation.
4. No non-LLM baselines, e.g., GNN embedding + LLMs.

**Questions:**

N/A

---

### Official Review · Reviewer_nkHu · 2025-11-01

**Soundness:** 3
**Presentation:** 3
**Contribution:** 3
**Rating:** 4
**Confidence:** 4

**Summary:**

The paper introduces GraphARC, a comprehensive and scalable new benchmark designed to evaluate abstract relational reasoning on graph-structured data. Generalizing the few-shot learning paradigm of the original grid-based Abstraction and Reasoning Corpus (ARC) , GraphARC tasks require a model to infer a transformation rule from 2-3 input-output graph pairs and apply it to a new test graph. The benchmark includes 21 distinct transformations (both color-based and structural) and can be automatically generated at scale across diverse graph families and sizes, enabling systematic testing of generalization. In their evaluation of state-of-the-art language models, the authors identify key limitations, including a "comprehension-execution gap" (models understand graph properties but fail to generate the correct transformed graph) , significant performance degradation as graph size increases , and a "transformation bias" where capable models paradoxically over-apply inferred rules.

**Strengths:**

1. Novel and Necessary Benchmark

The paper addresses a clear gap in AI evaluation. While relational reasoning is a core part of intelligence, existing benchmarks like ARC are confined to grids. GraphARC provides a more general and direct benchmark for this capability by using graph-structured data, moving beyond spatial or visual cues to test purely structural understanding.

2. Scalability and Diversity of Tasks

A significant advantage over the original ARC is that GraphARC tasks can be automatically generated at scale. The framework supports diverse graph families (e.g., trees, bipartite, random graphs) and sizes (tested up to 250 nodes). This "virtually unlimited supply of instances" is crucial for robustly evaluating model generalization and identifying scaling limitations.

3. Rigorous Evaluation Methodology

The evaluation pipeline (shown in Figure 2)  is a key strength. It innovatively separates model evaluation into two pathways: the "Full Output Path" (testing end-to-end execution) and the "Question-Based Path" (testing comprehension). This thoughtful design allows the authors to disentangle understanding from execution, leading directly to their main finding of a "comprehension-execution gap".

4. Extensive and Solid Experiments

The authors conducted a large-scale evaluation, testing a wide range of modern language models (including the Qwen3 series, LLaMA, and multiple OpenAI reasoning models like o1-mini, o3-mini, o4-mini, and GPT 5). The analysis, based on over 125,000 responses across 21 different tasks and multiple encoding formats, provides a solid empirical basis for the paper's conclusions.

5. Clear and Insightful Findings

The paper delivers more than just a leaderboard. It identifies three specific and significant failure modes of current LLMs: the comprehension-execution gap, the scaling barriers where performance collapses on larger graphs, and the "transformation bias" where more capable models over-apply reasoning. These findings are concrete, well-supported by data, and provide clear directions for future research.

**Weaknesses:**

1. Limited Scope of Graph Types and Tasks

The authors acknowledge that the benchmark is restricted to 21 tasks on undirected and unweighted graphs. This excludes many common and complex graph structures like directed graphs (which introduce concepts like predecessors/successors), weighted graphs (testing shortest paths, min-cuts, etc.), and multigraphs. The reasoning challenges in these more complex structures remain unevaluated.

2. Exclusion of Graph-Native Models

The evaluation focuses entirely on Large Language Models (LLMs) processing textual representations of graphs (adjacency or incident lists). The paper explicitly states that Graph Foundation Models (GFMs) and Graph Neural Networks (GNNs) were not benchmarked because they "cannot be straightforwardly adapted". While justified, this omission means the paper cannot compare text-based reasoning with models that have graph-structural inductive biases built-in.

3. Lack of a Formal Human Baseline

The paper "did not conduct a formal human evaluation", citing visualization challenges and time constraints. While an informal test on graduate students is mentioned, a formal baseline is a key component of the original ARC benchmark. Without it, it is difficult to formally contextualize the difficulty of the tasks or determine if the AI's failure modes are also common pitfalls for humans.

4. Reliance on Simple Prompting Strategies

The authors admit they "employed only straightforward prompting strategies rather than more sophisticated approaches". The reasoning capabilities of LLMs are known to be highly sensitive to prompting. The observed "comprehension-execution gap" might be partially mitigated by more advanced techniques like complex Chain-of-Thought (CoT), self-correction, or allowing the model to use tools (e.g., a code interpreter) to execute graph operations.

5. Potential for Task Ambiguity

The benchmark relies on inferring a complex rule from only 2-3 examples. The authors note the importance of "task uniqueness" and state they try to make ambiguity "negligible," but this isn't a formal guarantee. The case study of GPT 5's failures shows that models can misinterpret a rule (e.g., "degree exactly 3" vs. "degree at least 3") even when the examples seem clear to the authors, suggesting that some tasks might be inherently ambiguous or under-specified.

**Questions:**

Is it possible to consider a wider variety of graph structures, such as directed and weighted graphs. This would allow for the creation of tasks that test more complex reasoning (e.g., finding shortest paths, identifying source/sink nodes, or reasoning about flow).

Would it be possible to adapt the GraphARC framework to benchmark GNNs and emerging GFMs? This would require a different input-output format but would provide a crucial comparison between models reasoning over text and models reasoning directly on graph structures.

Would it be possible to perform a formal human evaluation, even on a subset of the tasks and text-based representations? This would help to more clearly distinguish between tasks that are "hard" in an absolute sense versus those that are specifically hard for the reasoning patterns of current AI models.

---

### Note · Authors · 2025-11-19

**Comment:**

We thank the reviewers for their detailed and constructive feedback. We appreciate the positive remarks regarding the extensiveness of experiments, scalability of the benchmark, and the failure mode analysis.

At this time, we have decided to withdraw the paper to address several points raised in the reviews, including broadening the range of tasks and establishing a human baseline. We also plan to explore how the benchmark could interface with existing graph native models.

**Withdrawal Confirmation:**

I have read and agree with the venue's withdrawal policy on behalf of myself and my co-authors.